# A scoping review on technology applications in agricultural extension

**Zhihong Xu**[1]*, **Anjorin Ezekiel Adeyemi**[1], **Emily Catalan**[1], **Shuai Ma**[1], **Ashlynn Kogut**[2], **Cristina Guzman**[1]

1 Department of Agricultural Leadership, Education and Communications, Texas A&M University, College Station, Texas, United States of America, 2 Department of Teaching, Learning, and Culture, Texas A&M University College, Station, Texas, United States of America

☯ These authors contributed equally to this work.
* xuzhihong@tamu.edu

**Data Availability Statement:** All relevant data for this study is publicly available from the Texas Data Repository (https://doi.org/10.18738/T8/VNLOTC).

## Abstract

Agricultural extension plays a crucial role in disseminating knowledge, empowering farmers, and advancing agricultural development. The effectiveness of these roles can be greatly improved by integrating technology. These technologies, often grouped into two categories–agricultural technology and educational technology–work together to yield the best outcomes. While several studies have been conducted using technologies in agricultural extension programs, no previous reviews have solely examined the impact of these technologies in agricultural extension, and this leaves a significant knowledge gap especially for professionals in this field. For this scoping review, we searched the five most relevant, reliable, and comprehensive databases (CAB Abstracts (Ovid), AGRICOLA (EBSCO), ERIC (EBSCO), Education Source (EBSCO), and Web of Science Core Collection) for articles focused on the use of technology for training farmers in agricultural extension settings. Fifty-four studies published between 2000 and 2022 on the use of technology in agricultural extension programs were included in this review. Our findings show that: (1) most studies were conducted in the last seven years (2016–2022) in the field of agronomy, with India being the most frequent country and Africa being the most notable region for the studies; (2) the quantitative research method was the most employed, while most of the included studies used more than one data collection approach; (3) multimedia was the most widely used educational technology, while most of the studies combined more than one agricultural technology such as pest and disease control, crop cultivation and harvesting practices; (4) the impacts of technology in agricultural extension were mostly mixed, while only the educational technology type had a statistically significant effect or impact of the intervention outcome. From an analysis of the results, we identified potential limitations in included studies' methodology and reporting that should be considered in the future like the need to further analyze the specific interactions between the two technology types and their impacts of some aspects of agricultural extension. We also looked at the characteristics of interventions, the impact of technology on agricultural extension programs, and current and future trends. We emphasized the gaps in the literature that need to be addressed.

**Funding:** The author(s) received no specific funding for this work.

**Competing interests:** The authors have declared that no competing interests exist.

## 1. Introduction

Agricultural extension programs play a crucial role in disseminating knowledge, empowering farmers, and driving agricultural development. From the earliest times, agricultural extension has been noted to traditionally, through the research scientists, develop products and methods which are transferred to the farmers through the extension agents for adoption. The transfer process which was mostly in-person or through radio communications [1] became largely inadequate to catch up with the expanding population as well as the rapid pace of development. This was further compounded by reduced government funding, uncertainties of the effectiveness of the methods, the extent of the relevance of the knowledge disseminated, and the appropriateness of the models [2] giving rise to introspection for paradigm shifts in the extension methods and practices. Hence, to enhance the effectiveness of these extension programs, the use of technology such as information and communication technologies (ICTs), digital technologies, farm simulation, and others became very much necessary. Rajkhowa & Qaim [3] noted that technology application has the potentials for improving the delivery of agricultural extension programs and disseminating agricultural research to farmers and producers since they can lower communication costs, improving smallholder market access and household welfare. By leveraging technology, agricultural extension can overcome geographical barriers, reach a wider audience, and provide access to valuable information and resources, leading to improved farming practices, increased productivity, and enhanced agricultural outcomes [4].

By exploring the application of technology in agricultural extension programs, this scoping review aims to shed light on the current state of research, identify gaps, and map the overall landscape of this rapidly evolving field. By examining journal articles, conference proceedings, and dissertations, this review specifically describes the outcomes of technology application in agricultural extension under three objectives which are the substantive features, methodological features, and characteristics of technology application in the context of agricultural extension. The findings of this scoping review will provide valuable insights for policymakers who are faced with the decision of expending their resources on the most effective yet economical technology. It can also provide researchers with empirical evidence supporting their decisions when designing adoption and diffusions models for agricultural innovations, as well as practitioners in the field of agricultural extension who will come face-to-face with the users of these innovations. The review will facilitate evidence-based decision-making and inform the development of effective policies and practices by offering an overview of the impact of technology application in agricultural extension. Moreover, it will foster collaboration among stakeholders, encouraging partnerships and knowledge-sharing to drive agricultural development.

Furthermore, the findings of this research will make significant contributions to establish a foundation for future studies. Through this study, we envisage a knowledge synthesis from included studies that could lead to a better understanding of the types, usage and effectiveness of the technology used in agricultural extension. By synthesizing existing knowledge, the review will identify areas where additional research is needed, thereby paving the way for further exploration and discovery. This contribution will advance the application of technology in agricultural extension and shape the future direction of the rapidly evolving field, ultimately leading to improved agricultural outcomes and sustainable development in farming communities worldwide.

## 2. Literature review

### 2.1. Agricultural extension

Throughout the history of agricultural extension, there have been a variety of definitions of agricultural extension based on who is involved, the location, and the method used. For example, Msuya et al. [5] described agricultural extension as a way for small farmers to access new technologies, while Birkhaeuser et al. [6] viewed it as a common form of public-sector support for spreading knowledge. Rivera et al. [7] (5) on the other hand explained how agricultural extension serves as a link to increase productivity and efficiency for farmers and researchers and makes it easier to share innovations among farmers. Of all these, however, the most often cited is Maunder's [8] comprehensive definition where agricultural extension was defined as "a service or system which assists farm people, through educational procedures, in improving farming methods and techniques, increasing production efficiency and income, bettering their levels of living, and lifting the social and educational standards of rural life." From these definitions, to achieve its goals, agricultural extension must incorporate key components such as farmers and/or farming households, knowledge diffusion/education, and willingness to change on the part of the farmer.

This scoping review will align with the definition given by Maunder [8] within the framework that the studies included involve farmers and/or farming households with the aim of mobilizing resources towards their farming objectives.

### 2.2. Technology application in agricultural extension

The importance of technology in enhancing agricultural productivity cannot be overstated, and agricultural extension plays a crucial role in achieving this objective. Technology, with its innovative tools and applications, has been identified as a game-changer in the agricultural sector [9]. It has revolutionized farming practices, empowered farmers to increase productivity [10–12], optimized resource utilization [13,14], and addressed sustainability challenges [15–17].

Technology application (TA) in agriculture has been extensively explored from two distinct yet interconnected perspectives. The first viewpoint focuses on the use of technology/innovation as a factor or component of production. Studies falling under this theme investigate aspects such as improved seed varieties [18–20]; farm machinery, including tractors, plows, harvesters, and similar equipment [21,22]; drones, animal trackers [23] and more recently robots [24]; and the Internet of Things (IoT) [25]. These studies perceive these technologies as resources that are consumed or incorporated into the farming system, recognizing that their absence may hinder one or more crucial stages of the production process.

The second perspective regards technology in agriculture primarily as a means of enhancing knowledge transfer and skills development, often referred to as educational technology (ET). These studies, which often focus on technology-enabled information dissemination, training, and capacity building, incorporate technologies such as virtual reality (VR) and augmented reality (AR), Information and Communication Technology (ICT) [26–29], smartphones/mobile applications [30–32], online platforms and websites [33,34], e-learning and webinars [35–37], and social media and online communities [38–40].

In this scoping review, we categorize the TA in agricultural extension into two groups: 1) agricultural technologies/innovations used during production and 2) educational technologies employed for training and facilitating the adoption of these agricultural technologies.

By integrating ET tools such as videos, smartphones, online training, and tablets, agricultural extension services/agents can significantly enhance the effectiveness of information transfer while reducing costs. This approach helps farmers in remote areas easily access timely

information, such as weather variables and market factors. Studies have demonstrated the efficacy of these tools, including videos, smartphones, and tablets, in improving agricultural practices among farmers [38–41].

The potential of combining ET and agricultural technology/innovations is highly promising. However, a comprehensive review of previous studies to ascertain the practical outcomes is still lacking. By examining the existing literature, this scoping review aims to bridge the gap in understanding the practical implications of integrating ET and agricultural technology/innovations (ATI). The findings of this study will shed light on the effectiveness and impact of these combined approaches in agricultural extension services.

### 2.3. Previous studies and research gap

While previous studies have explored agricultural extension and TA separately [42], there is a lack of examination of the relationship between these two topics. This scoping review aims to address this gap by examining them simultaneously. Existing literature reviews have touched upon related aspects, such as the role of agricultural extension in the transfer and adoption of AT [43] and the use of ICT for agricultural extension in developing countries [44]. However, these prior reviews do not fully examine the relationship between agricultural extension and TA. Altalb et al. [43] highlight the importance of agricultural extension in the development of the agricultural sector and how it aids in transferring necessary knowledge to farmers. Although Altalb et al. [43] discussed various technologies and innovations in the agricultural sector, their objective was to explore how agricultural extension could transfer that information to the farmers. In contrast, our study focuses on not just information transfer but goes ahead to examine the extension system.

Aker [44] highlights the need to adopt better AT like fertilizer, seeds, and other farming methods in developing countries and the potential of technology mechanisms, such as voice, text, internet, and mobile phone, to reach farmers and enhance knowledge, ultimately leading to an increase in the economy. However, the study did not delve into the direct TA within agricultural extension; and failed to provide examples or results that demonstrate how technology can be implemented through agricultural extension.

By addressing these gaps and incorporating potential recommendations derived from a comprehensive analysis of previous studies, this scoping review aims to contribute to enhancing productivity and bridging the divide between TA and agricultural extension practices by providing empirical evidence of amongst other things, the impact technology can make in agricultural extension.

## 3. Research questions

This present scoping review aims to investigate the effect of technology application on agricultural extension by examining existing empirical studies. The study focuses on analyzing the substantive features, methodological features, and characteristics of technology application in the context of agricultural extension. The research questions guiding this study are as follows:

- What are the substantive features of the included studies, including publication information (year of publication and journal name), country/region information, and agricultural field?

- What are the methodological features of the included studies, such as the research methods employed, data collection approaches, and sample size?

- What are the characteristics of the technology used in agricultural extension, including the type of educational technology, agricultural technology, and the overall effect of technology on agricultural extension?

## 4. Research method

### 4.1. Search strategies

To comprehensively search for studies, we searched five databases: CAB Abstracts (Ovid), AGRICOLA (EBSCO), ERIC (EBSCO), Education Source (EBSCO), and Web of Science Core Collection. These databases cover literature in the agriculture, applied life sciences, and education disciplines. The database search was developed in CAB Abstracts and run on October 28, 2022. The original search was modified for the additional databases, and the additional databases were searched on November 1, 2022. The search consisted of subject terms and keywords related to the two core concepts of educational technology and agricultural extension. Keywords were searched for in the title and abstract fields. The full search strategies for CAB Abstracts and the other four databases can be found in Appendix A in S2 File.

### 4.2. Inclusion and exclusion criteria

This scoping review used specific inclusion and exclusion criteria to identify studies examining the impact of technology application on agricultural extension.

1. The included studies must have examined the effect of technology application on agricultural extension. Articles were excluded if they were not about technology, were not within the agricultural extension context, and did not examine the effect of technology on agricultural extension.

2. Technologies for this study were defined as the educational technology such as multimedia, smartphones, iPads and tablets, digital simulation devices and others used by the agricultural extension services/specialists/agents to facilitate educational training under which knowledge, skills, and content are transferred in the form of agricultural technology/innovations such as seed planting knowledge, disease and pest prevention practices, improved varieties, record keeping and others to the farmers and other stakeholders in an agricultural extension setting. Unless agricultural technology also qualifies as an educational technology (e.g., GPS), such studies were excluded.

3. Included studies must have been conducted under the context of agricultural extension programs, which take place in an informal, out-of-school setting; directly involve farmers and/or farming households; and pertain to their farming enterprises.

4. Included studies must report detailed information on the effect of technology on agricultural extension, which should include the sample size, experimental design, and detailed results (either quantitative or qualitative). Conference abstracts on this topic will be excluded.

5. Included studies must have reported an assessment of technology's impact/effect on agricultural extension, qualitatively or quantitatively, such as an empirical study (intervention or case studies). Articles that generally discuss the trends or the importance of technology in agricultural extension were excluded.

6. The included studies must have been published in a journal, as a conference proceeding, or policy paper from January 1, 2000, to November 1, 2022, and available in English. We selected this period to ensure that we covered the latest studies and documented the rapid progressions of technology in agricultural extension since 2000 [45,46]. Secondary data analysis, literature reviews, book reviews, book chapters, and reports were excluded.

### 4.3. Coding scheme

To ensure efficient data extraction and analysis, a comprehensive coding system was developed to categorize and organize the information from the included studies. The coding system facilitated the examination of substantive and methodological features of the studies, specifically focusing on the impact of technology application on agricultural extension. Sub-categories were created within the coding system to distinguish between agricultural technologies and educational technologies, enabling a detailed analysis of the key features of each. This coding played a crucial role in understanding and interpreting the findings of the included studies.

### 4.3.1 Substantive features of the studies

The substantive features of the studies included their publication information, geographic location (country/region), and the included studies' agricultural field/enterprise concentrations. Our primary objective was to comprehensively analyze publication patterns within the discipline. We aimed to identify journals that had a significant impact based on their titles and publication dates. Furthermore, we sought to compare agricultural extension technology trends across various countries and regions.

We categorized the agricultural fields/enterprises in which educational technologies were predominantly utilized. The coding scheme (Table 1) classified the agricultural fields/enterprise as follows: agricultural economics, including food processing such as making raisins and any value-addition processing; agricultural engineering, including mechanization; agronomy, encompassing crop production and other crop-related enterprises; animal husbandry, incorporating animal production, fisheries, and other livestock-related enterprises; and mixed when the agricultural field/enterprise included more than one.

**4.3.2 Methodological features of the studies.** The methodological features of the included studies encompassed several aspects, including the research methods employed, data collection approaches, the use of inferential statistics, and units of sample size. These components were examined to gain insights into the study design and methodology employed in investigating the impact of technology on agricultural extension.

The research methods were grouped into quantitative, qualitative, and mixed methods. Quantitative studies used descriptive and inferential statistics, while qualitative studies followed Denzin & Lincoln's [47] definition of interpretive practices across different disciplines. We categorized the research methods into quantitative and qualitative because these are the primary categories of educational research. Since studies use both quantitative and qualitative methods, we categorized mixed methods studies as those studies that used both quantitative and qualitative approaches to collect and analyze data.

The data collection approaches included surveys, questionnaires, interviews, focus group discussions (FGD), and assessments. If the study applied more than one data collection

**Table 1. Breakdown of identified agricultural field/enterprise.**

| Agricultural field/enterprise | Field Content |
| --- | --- |
| Agricultural economics | Food processing (making raisins)/ value-addition |
| Agricultural engineering | Mechanization |
| Agronomy | Crop production, castor cultivation |
| Animal husbandry | Animal production, fisheries, apiculture |
| mixed | |

approach, it was coded as a mixed method. We also documented whether the researchers employed inferential statistics to examine the impact of educational technology on agricultural extension.

We also considered the sample size units for the selected studies. The sample units were varied, making it difficult to unify the sample sizes. Therefore, we coded the sample size units as households, individuals, and villages, and in studies that used more than one unit, we coded them as mixed.

**4.3.3 Characteristics of technology in agricultural extension.** We categorized the characteristics of the technology applications used in agricultural extension. The types of educational technologies were coded under the following categories: multimedia (video, audio, photographs, video animation, radio); mobile apps/smartphones; online/web-based; digital simulations; and mixed for those studies that used more than one.

We distinguished between ET and an AT/I were being transferred to the farmers. We categorized the agricultural technology/innovation into various groups: crop cultivation/harvesting practices, product processing, pest and disease control, and knowledge/skill/general agricultural education. The first category was crop cultivation/harvesting practices including spacing and fertilizer application, castor cultivation, cotton production, protection technology, rice intensification system, integrated soil fertility management, soil modules, and sugarcane ratoon management practices. Another category pertained to product processing, specifically the storage of beans in jerrycans. Furthermore, we grouped pest and disease control methods such as the use of neem as an insecticide, disease management, weed control practices, and the management of Fall armyworms. Knowledge/skill/general agricultural education was another category, including topics like insurance advisory, record keeping, knowledge sharing and joint decision making, climate adaptation strategies, and practices. In cases where multiple agricultural technologies/innovations were identified, they were classified under a mixed category.

For characteristics of the intervention, we coded the duration (how long) and the intensity (how often) of the technology intervention as well as the timing of the measurement of the impact /effect. For the duration and intensity of the intervention, we considered how many studies provided the information and reported how they were reported. The interval between intervention and measurement of effect was coded as immediate, short-term, long-term, mixed, and unspecified for those studies that did not clearly state the timing for the measurement. Additionally, we coded whether the use of technologies had a positive, negative, non-significant, or mixed impact and whether the effect sizes were reported.

## 4.4. Data collection and data analysis

To identify eligible studies, we followed the screening process illustrated in Fig 1. After removing duplicates, we screened 4,170 unique references for eligibility. The research team screened the article titles and abstracts using the inclusion/exclusion criteria. After the first round of screening, 69.71% (2,808 articles) were excluded. After an initial screening, the authors reviewed the full text of 1,319 articles. Out of these, 61 articles were found to be eligible for inclusion in the review. During the coding process, seven articles were excluded for different reasons. Finally, 54 articles were included in the final coding stage.

The coding scheme was created by the first two authors, who independently coded a set of 20 randomly selected articles. Subsequently, the coding scheme was employed by the first five authors to code the articles using Microsoft Excel. As noted by Belur et al. [48], the interrater reliability (IRR) of a good systematic review strengthens the transparency and replicability of the process leading to the results from such reviews. Thus, to calibrate the coding, the 54

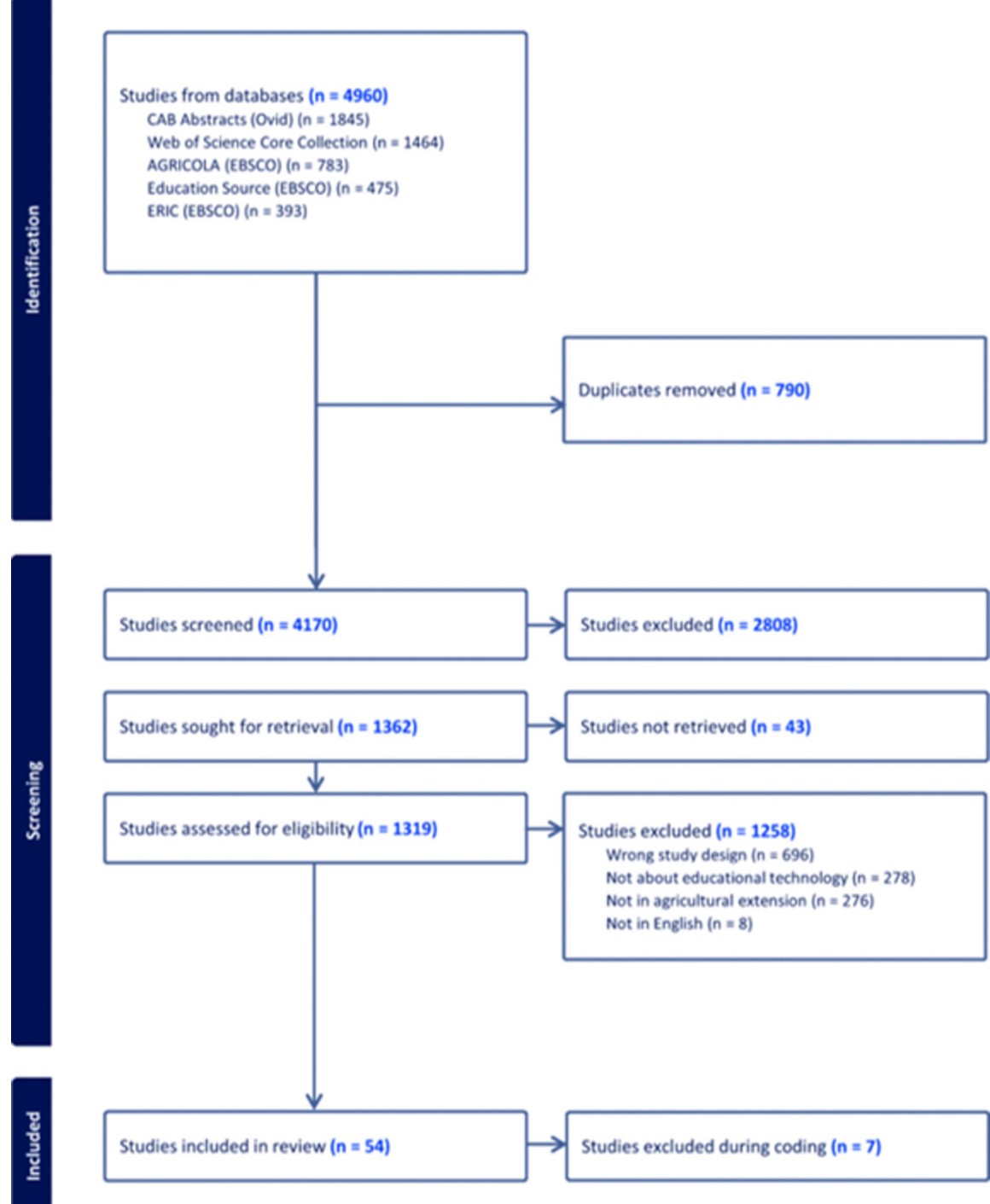

**Fig 1. PRISMA flow diagram of the search and screening process.**

articles were initially coded independently, resulting in an initial round of IRR of 81.30% which was calculated by the percentage of agreement between the coders. In case of conflicts, the first author acted as the arbiter and resolved the discrepancies. Eventually, a unanimous

agreement was achieved regarding the coding of the articles. Descriptive statistical analyses were conducted to address our research questions.

For the data analysis, simple descriptive statistics such as frequencies, percentages, charts, and graphs were used to analyze and present the results for an understanding of the substantive and methodological features of the studies. For the characteristics of technology in agricultural extension, we conducted a crosstabulation and Chi-square analyses of the type of educational and agricultural technology used on the effect/impact of the intervention.

# 5. Results and discussion

## 5.1. Substantive features of the studies

**Publication information.**   Among the 54 included studies, a noteworthy observation was the concentrated distribution of publications within the past six years (2016–2022). As depicted in Fig 2, two studies (3.70%) were published from 2001–2005. Six studies (11.11%) were published from 2006 to 2010; eight studies (14.81%) were published from 2011–2015. The majority of publications, comprising 38 studies (70.38%), were published between 2016 and 2022. This trend indicates a significant increase in research activity from 2001 to 2022, with a surge in studies focusing on educational technology after 2016. The rapid development and adoption of various training platforms for farmers accentuated by the global impact of the Covid-19 pandemic, has underscored the pressing need for technology-assisted agricultural extension [49–51].

Out of the 54 studies examined, the majority (n = 49; 90.74%) were published in journals. Policy/discussion papers constituted 7.41% (n = 4) of the studies, while conference papers represented 1.85% (n = 1). Notably, no theses nor dissertations were included in the present scoping review. The absence of such works highlights a potential avenue for graduate students to explore the intersection of educational technology and agricultural extension. Among the journals, 61.22% (n = 30) appeared only once, while the remaining 38.78% (n = 19) published multiple articles included in this review. Details on the number of articles per journal can be found in Table 2. The journals that published the most papers in the field are: The Journal of Agricultural Education and Extension (JAEE) (n = 4), JIAEE (Journal of International Agricultural and Extension Education) (n = 3), International Journal of Agricultural Sustainability (n = 3), Information Technology for Development (n = 3), International Food Policy Research Institute (n = 3).

**Country.**   The included studies encompassed a diverse range of countries, with notable concentrations in India, Uganda, Benin, and the U.S.A. Overall, research were conducted in 17 different countries. India accounted for 29.63% (n = 16) of the studies, while Uganda represented 16.67% (n = 9). Both Benin and the U.S.A. had five studies (9.25%) conducted in each country. Kenya accounted for 7.40% (n = 4) of the studies, while Mali, Ethiopia, and Bangladesh each had two studies (3.70%) conducted in each of these countries. The other 16.67% (n = 9) of the studies were conducted in Nigeria, Mozambique, Malawi, Bolivia, Ghana, France, Senegal, China, and Burkina Faso, with one study in each country respectively. The prevalence of studies conducted in the predominantly developing countries (excluding USA), is indicative of the high number of farm families in these regions compared to extension services. Technology therefore plays a crucial role in bridging the gap in effectively reaching a large population of farmers in these areas within a short period [52–54].

**Region.**   The research primarily focused on the regions of Africa, Asia, and North America. Out of 54 studies analyzed, 28 studies (51.85%) were conducted in Africa, while 19 studies (35.19%) were carried out in Asia. Additionally, five studies (9.26%) were conducted in North America, only one study (1.85%) was conducted in South America, and one study (1.85%) was

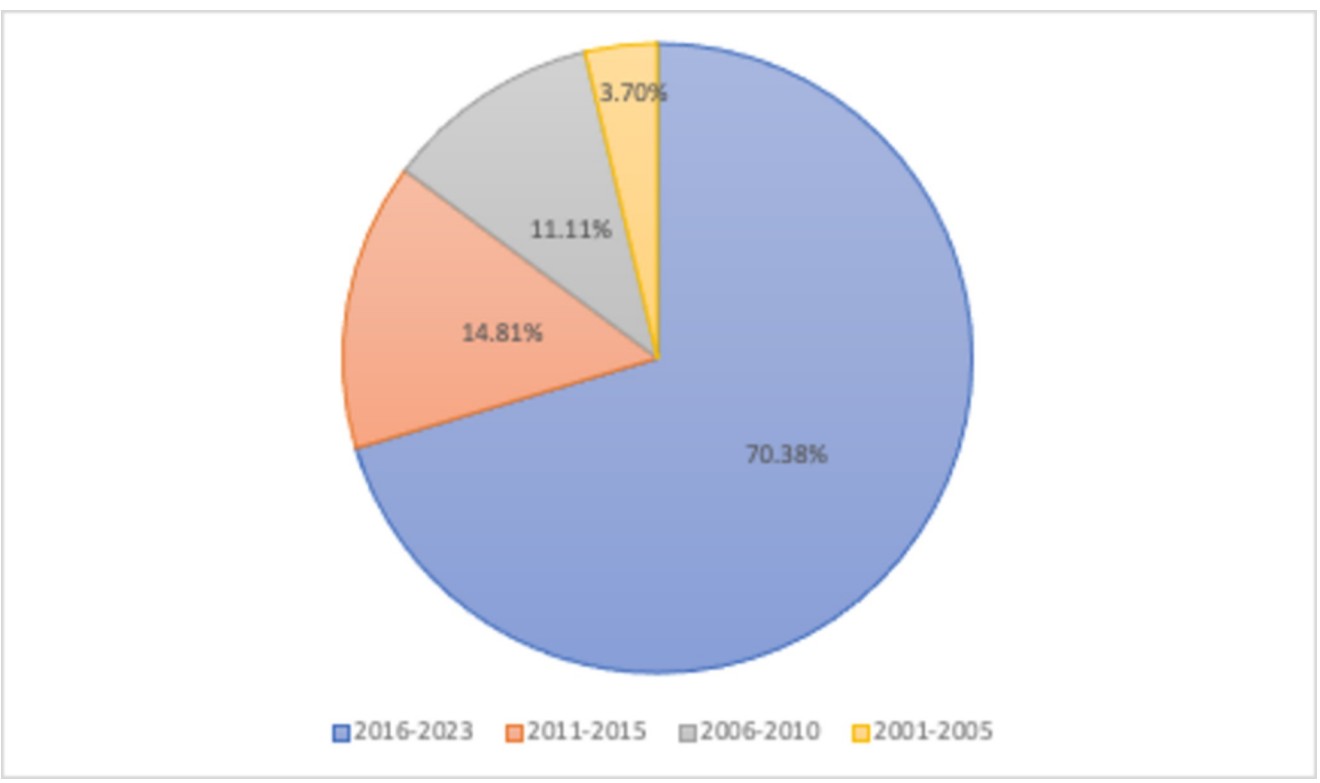

**Fig 2. Distribution of included articles by publication information.**

**Table 2. Journals, conferences, and policy/discussion paper of included articles.**

| Journal | Number of articles included in the review |
| --- | --- |
| The Journal of Agricultural Education and Extension (JAEE) | 4 |
| JIAEE (Journal of International Agricultural and Extension Education) | 3 |
| International Journal of Agricultural Sustainability | 3 |
| Information Technology for Development | 3 |
| PLOS One | 2 |
| Journal of Extension | 2 |
| Journal of Agricultural and Food Information | 2 |
| Other journals (e.g., Journal of Development Effectiveness, Development in Practice, and Journal of Plant Development Sciences) | 30* |
| Conference | Number of articles included in the review |
| Extnicon 2018 | 1 |
| Policy/discussion paper | Number of articles included in the review |
| International Food Policy Research Institute | 3 |
| Economic Development and Cultural Change | 1 |

Note: *other articles only appear once but in separate journals.

conducted in Europe. Notably, no studies specifically targeted Antarctica or Australia/Oceania. These findings highlight the active contributions of Africa, Asia, and North America to research in the field of educational technology in agricultural extension. However, the dearth of research from Australia/Oceania and Europe in our included studies suggests a need for further investigation in these regions. For instance, Australia/Oceania, renowned for its expertise in animal husbandry due to the combination of large land areas, a substantial livestock population but relatively limited investment in infrastructure and human resources [55], presents a particularly interesting area for future researchers to explore.

**Agricultural field.** The majority of the included studies exhibited a strong focus on agronomy. As shown in Fig 3, 43 studies (79.63%) were centered around agronomy. Additionally, six studies (11.11%) pertained to animal husbandry, three studies (5.56%) involved a mixed focus, and two studies (3.70%) were related to agricultural economics. The imbalance in the distribution of studies suggests a potential opportunity to explore and utilize educational technology in fields such as animal husbandry, agricultural economics and engineering, and other mixed areas. By expanding the application of technology to these underrepresented domains, a more comprehensive and inclusive approach can be adopted within the agricultural extension.

## 5.2. Methodological features of the studies

**Research methods.** The analysis of the research methods employed in the included studies indicated a predominant use of the quantitative research method. Thirty-seven studies (68.52%) utilized the quantitative method, while 14 studies (25.93%) employed a mixed-method approach. Only three studies (5.55%) used the qualitative method. These findings align with a scoping review (authors, under review) on educational technology in agricultural education, which also observed a prevalence of quantitative and mixed methods research as the commonly adopted approaches in this field. The rationale behind the prevalent use quantitative research methods may stem from several factors. Firstly, researchers may have already recognized the importance and advantages of educational technology in the agricultural extension field, largely owing to the extensive body of research on the of educational technology in general education. Consequently, their inclination might have been to substantiate their existing hypotheses within the extension field. It is worth noting that quantitative research often leans towards a confirmatory and deductive approach, in contrast to the more exploratory nature often associated qualitative research [56]. Additionally, another possible reason for favoring quantitative methods could be attributed to the inherent limitations of qualitative methods. Qualitative findings are typically context-specific and may not readily generalize to a broader population [56].

We recommended that researchers employ more mixed methods research designs, which combine both quantitative and qualitative approaches, because it offers additional advantages in social science research [57]. For example, mixed methods research allows researchers to obtain a more comprehensive understanding of complex social phenomena by integrating numerical data with in-depth qualitative insights. The predominant use of mixed research methods in social sciences research is driven by the need for empirical evidence, objectivity, generalizability, and the ability to establish causal relationships and test theories [57,58].

**Data collection approaches.** The analysis of data collection approaches revealed that mixed approaches were the most utilized among the included studies. Out of the 54 studies, 19 studies (35.19%) used mixed approaches, 13 studies (24.07%) relied on assessments as their primary data collection approach, and 11 studies (20.37%) utilized surveys. Additionally, eight studies (14.81%) used interviews, two studies (3.70%) employed questionnaires, and one study

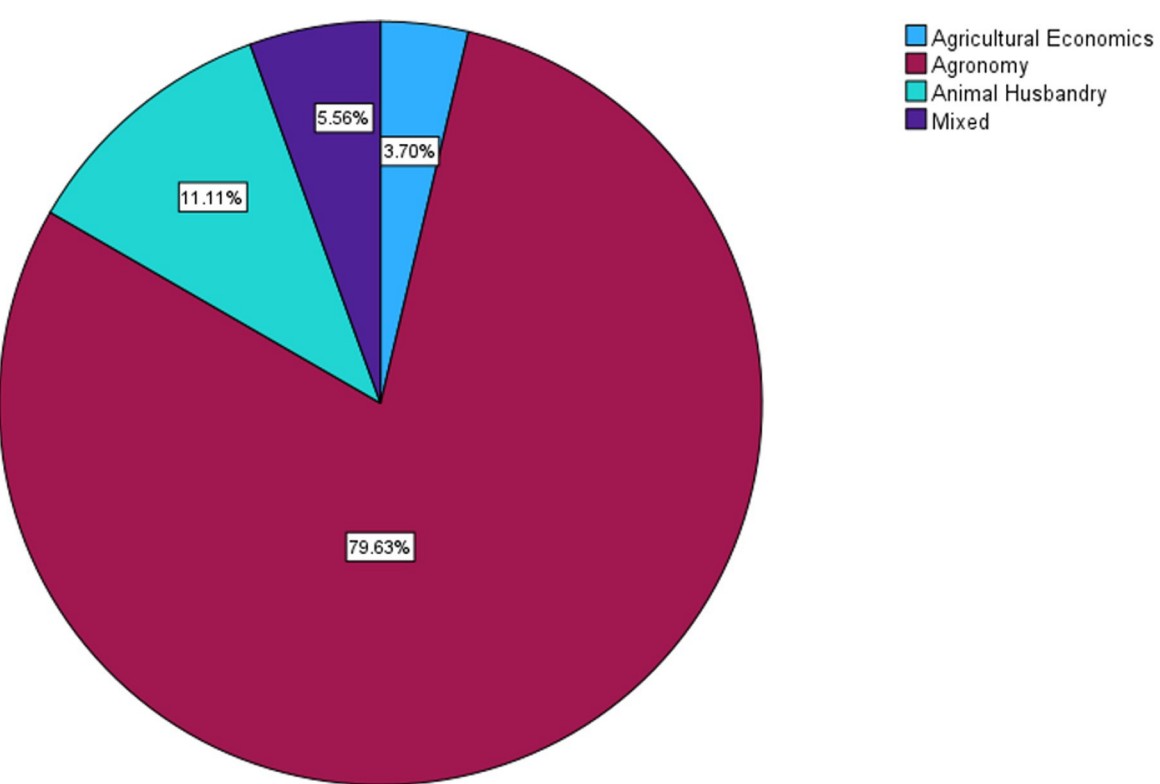

**Fig 3. Distribution of included articles by agricultural fields.**

(1.86%) did not specify the data collection method used. This diversity in data collection methods highlights the importance of employing a range of approaches to gather comprehensive and nuanced information within the field of educational technology in agricultural extension.

**Inferential statistics.** The analysis of inferential statistics showed that a majority of the studies included employed this statistical approach. Among the 54 included studies, 70.37% ($n = 38$) of the studies utilized inferential statistics to analyze their data. On the other hand, 29.63% ($n = 16$) of the studies did not use inferential statistics in their data analysis. The prevalent use of inferential statistics reflects the researchers' intention to make inferences and draw broader conclusions about the relationship between educational technology and agricultural extension based on their data.

**Unit of sample size.** The included studies employed a variety of units for reporting sample size, with individuals being the most prevalent sample size unit. Out of the 54 included studies, 41 studies (75.94%) used individuals as the sample size unit. Additionally, six studies (11.11%) used households, four studies (7.40%) employed mixed units, one study (1.85%) used villages, and two studies (3.70%) did not report the sample size unit. The diversity in sample size units may be attributed to the specific characteristics of the agricultural field and the grouping involved, such as considering households or villages as a whole when studying agricultural practices. In future research endeavors, it would be beneficial to adopt a diverse array of sample size units, given the intricacy and distinctiveness of the agricultural extension field. Furthermore, there is room for investigation into the effectiveness of employing various sample size units. It is worth considering that social interaction within the households, villages, or communities within the group might be a significant factor contributing to the learning outcomes, in addition to individual interactions with technology. To gain a deeper understanding

of this aspect, both quantitative or qualitative research approaches can be employed to explore the dynamics of human interaction within a shared learning community in the context of agricultural extension.

Among the 41 studies that employed individuals as the sample size unit, we adhered to the commonly used quantitative research guidelines: studies with less than 100 participants were considered small samples, studies between 100–250 participants were classified as medium samples, and studies with over 250 participants were considered large samples [59,60]. The sample size for studies using individuals as the unit ranged from 6 to 58872 participants. Among these studies, 58.54% ($n = 24$) of the studies had a medium sample size, 24.39% ($n = 10$) of the studies had a small sample size, and 17.07% ($n = 7$) of the studies had a large sample size. Our findings suggest that most studies used a medium sample size when using individuals as the sample size unit. However, specific studies focusing on ET in educational settings suggested a prevalence of small sample size studies (60). This divergence could be attributed to contextual variations, particularly since agricultural extension studies typically involve a larger number of participants.

## 5.3. Characteristics of technology in agricultural extension

**Educational technology.** In our review of 54 studies, we discovered the utilization of various ET in agricultural extension. As shown in Fig 4, multimedia emerged as the most frequently used ($n = 27$, 50.00%), followed by studies that incorporated multiple types ($n = 15$, 27.78%). Additionally, mobile apps/smartphones were used in nine studies (16.67%), online/web-based applications in two studies (3.70%), and digital games/simulations in only one study (1.85%).

These findings differ from a similar review conducted on the use of ET in agricultural education by Xu et al. [61] Among the 83 included studies in their review, they found that the most used ET was online/distance education, followed by simulation/digital games and then, multimedia and traditional technology. This stark contrast may be attributable to the different contexts or settings in which agricultural education and agricultural extension are practiced. Agricultural education primarily takes place within formal educational institutions, involving students, academics, and professionals with higher levels of academic qualifications. On the other hand, agricultural extension often occurs in non-formal settings, predominantly involving farmers who may have varying levels of academic attainment. This is further supported by Mwololo et al.'s [62] finding that socio-economic factors such as age, education, and gender influenced farmers' preference for agricultural extension methods, specifically farmers' field schools (FFS), farmer to farmer (F2F), or mass media. In addition, the role and characteristic of multimedia contributed to the most frequent use as ET for farmers in the extension field. Multimedia plays an important role in agricultural extension serving as the most powerful opinion maker in this information era, and can help transfer agricultural information [63]. Multimedia is simple, direct, and intuitive in nature thereby making it very comprehensive for farmers who have limited educational level and technology literacy to attain knowledge and skills competency. The majority of our included studies were conducted in Africa and Asia with representative countries like India and Nigeria. In developing countries, farmers' educational level and current technology literacy remains limited due to the lag of development of the whole country economically, socially and technology and limited funding opportunities/resources for further improvement. Simple and cost-effective ET like multimedia would be preferred compared to complex ones.

Among the various forms of ET used in agricultural extension, video or video-mediated extension emerged as the most prominent. Horner et al. [64] conducted an experimental study

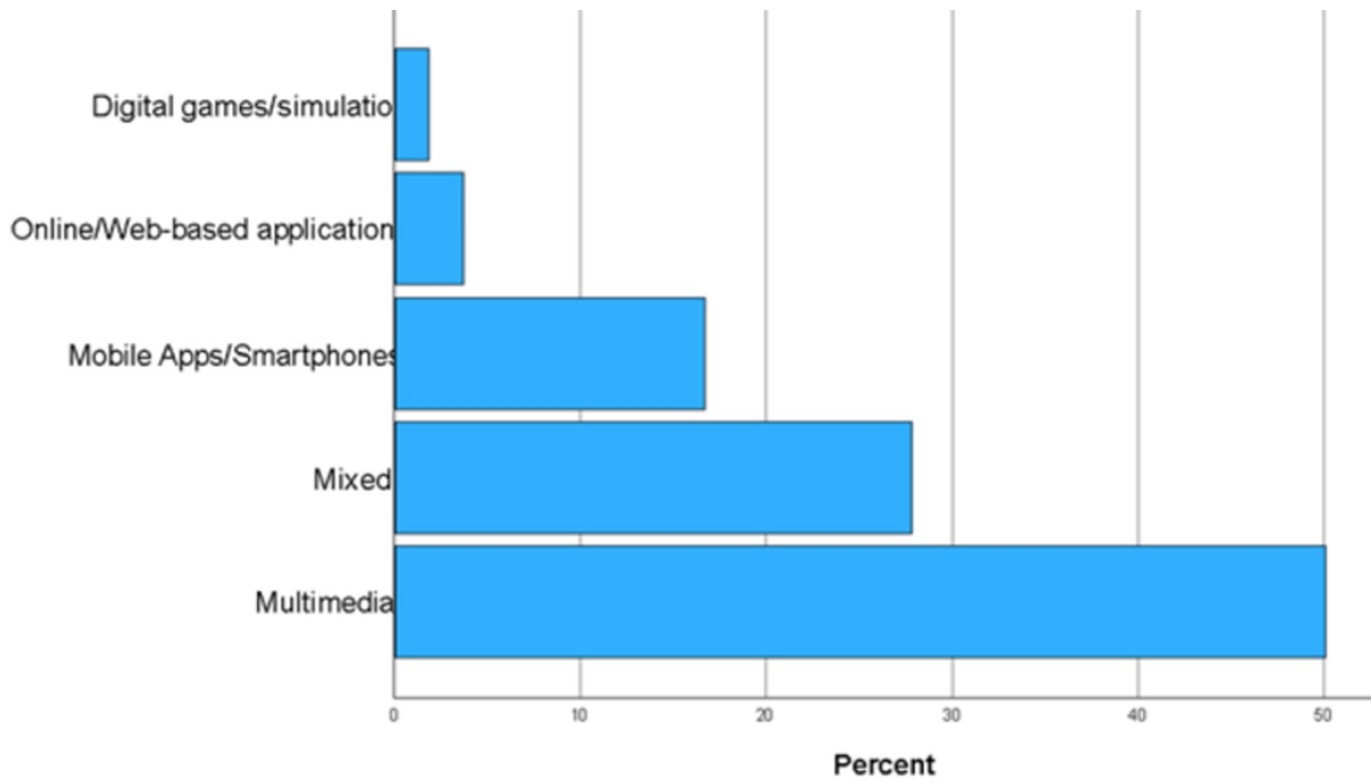

**Fig 4. Distribution of publications according to the type of educational technology.**

in Ethiopia to assess the effectiveness of video-based extension. They compared traditional agricultural extension methods with the incorporation of videos and found that the latter was more effective in increasing farmers' knowledge and adoption of complex agricultural technologies such as composting, blended fertilizer, improved seeds, line seeding, and lime. Chowdhury et al. [65] conducted a study in Bangladesh focusing on enhancing farmers' capacity for botanical pesticide innovation through video-mediated learning. They observed a significant increase in knowledge about botanical pesticides in both male and female farmers who participated in the video-mediated group. Several other studies [38,66–70] have also incorporated video-based multimedia in their agricultural extension programs.

The prevalence of video-mediated extension in agricultural extension programs underscores its effectiveness in delivering information and promoting knowledge acquisition among farmers. By utilizing videos, extension practitioners can visually demonstrate agricultural techniques, showcase best practices, and present success stories, thereby enhancing farmers' understanding and motivation to adopt agricultural practices. This multimedia approach is particularly beneficial in non-formal settings where farmers may have varying levels of education and diverse learning preferences.

In our analysis of 54 articles exploring the use of educational technology for transmitting agricultural technology/innovation to farmers, we identified multiple themes in the types of agricultural technologies. Most of the articles ($n = 21$, 38.89%) discussed a combination of agricultural technologies, indicating a mixed approach. Pest/disease control technology was the next most used agricultural technology ($n = 11$, 20.37%). Another 10 articles (18.52%) focused on crop cultivation/harvesting practices, six articles (11.11%) covered product processing

technology, and the remaining six articles (11.11%) focused on knowledge/skill/general agricultural education.

The agricultural technology and innovations covered in our included studies varied. Some studies incorporated a combination of technologies like row planting, precise seeding rates, and urea dressing [68]; tillage and sowing machinery [71], planting methods, weeding and fertilizer application [72,73]; identifying growth stages and improving yield predictions [74]; and seed selection, storage and handling [67].

Several studies also examined technologies and innovations for controlling pests and diseases. For instance, Chowdhry et al. [65] explored the use of botanical pesticides, Bentley et al. [75] investigated methods for controlling bacterial wilt (BW) in potatoes, and Dione et al. [76] focused on biosecurity messages for managing African swine fever. Other studies have been conducted on crop cultivation and harvesting practices. Dechamma et al. [77] studied the production practices of tomato crops, and Ding et al. [78] focused on nitrogen management practices in crop production. Additionally, Bello-Bravo et al. [79] and Sidam et al. [80] researched technologies related to product processing, such as storing beans in jerry cans and making raisins.

The last category of studies included those that focused on knowledge and skills/general agricultural education such as knowledge and awareness about agricultural credit [31], climate information [81], information about cattle handling [82], and backyard poultry farming [83].

**Intervention characteristics of technology.**   We classified the duration of the technology intervention, the intensity of the intervention, and the interval between the intervention and the measurement of its effect. Regarding the duration of the technology intervention, nine studies (16.68%) did not provide information on the duration. Eight studies (14.81%) implemented interventions that lasted less than a week, while seven studies (12.96%) had interventions that ranged from one week to 12 weeks (3 months). Eleven studies (20.37%) reported interventions lasting between 13 weeks to 24 weeks (6 months), while eight studies (14.81%) had interventions lasting between 25 weeks to 48 weeks (1 year). Furthermore, eleven studies (20.37%) documented interventions lasting from 48 weeks (1 year) to 192 weeks (4 years).

As for the intensity of the intervention, 64.81% ($n = 35$) of the studies did not provide information on the intensity, while 35.19% ($n = 19$) did include details on the intensity. Out of the 19 studies that reported the intensity of the intervention, six (31.58%) specified the frequency of the intervention, such as two sessions per week or two messages per week. Thirteen studies (68.42%) provided precise information on the exact time of each session or video of the intervention, which varied from two minutes to as long as two days. These findings indicate that a significant majority of studies should have included more detailed information on the intensity and duration of the intervention. As the intensity and duration are crucial components of an intervention, they play a significant role in interventions' effectiveness. Future research should place greater emphasis on exploring intensity and duration in greater depth and on detailed reporting of intervention components.

Regarding the interval between the intervention and the measurement of its effect, researchers exhibited a preference for measuring immediate effects, followed by long-term effects, short-term effects, and a mixed approach. Among the reviewed studies, 22 studies (40.74%) measured the immediate effect, 16 studies (29.64%) focused on the long-term effects (more than three months), seven studies (12.96%) assessed the short-term effects (within three months), two studies (3.70%) used a mixed interval between the intervention and the measurement of its effect, and seven studies (12.96%) did not specify the interval between the intervention and the measurement of its effects.

**Effect of technology application in agricultural extension.**   The effect or impact of using technologies in agricultural extension showed diverse outcomes across the 54 studies. Among

those studies, 35 articles (64.82%) recorded positive outcomes, while 15 articles (27.78%) documented mixed outcomes, suggesting a combination of positive and potentially less favorable results. Two articles (3.70%) reported non-significant outcomes, indicating that the technologies did not have a statistically significant impact on agricultural extension. Finally, the last two articles (3.70%) did not specify the outcomes achieved.

In one study with mixed outcomes, Bentley et al. [75] compared three agricultural extension methods (FFS, community workshops, and radio) for their effectiveness in teaching Bolivian farmers about BW of potato. Their findings found that while radio listeners received information about topics like diagnosing BW, crop sanitation practices, use of healthy seed, crop rotation, and incorporation of manure first from the radio, they never took any concrete action that led to the actual adoption of those agricultural technologies when compared to the FFS groups and the workshop attendees. So, while radio increased awareness about the AT, it fell short in the actual adoption.

Another study that reported mixed outcomes was that of Ding et al. [78] where ICT-based agricultural advisory services were used for nitrogen management in wheat production in China. The study sought to examine the effects of ICT-based extension services on the adoption of sustainable farming practices like nitrogen control in wheat production and found that while there was no reduction in the use of N-fertilizer for wheat production, the ICT-based services prompted farmers to adopt N-fertilizer use towards site-specific management. So, whereas the educational technology fell short of convincing the farmers to reduce their N-fertilizer usage in wheat production, it achieved the unintended goal of making the farmers adopt some site-specific management practices of N-usage.

In addition, we conducted cross-tabulation analyses and employed Chi-square tests to assess the associations between different types of educational technology, agricultural technology, and the resulting effects or impacts of the implemented technology interventions. Among the 54 articles, two articles did not specify the intervention effect.

Based on the findings presented in Table 3, a significant relationship was observed between the type of educational technology utilized and the resulting effect or impact of the intervention. The statistical analysis revealed a significant result of $\chi2$ $(8, n = 52) = 28.67, p < .001$, indicating that the type of educational technology employed influenced the outcomes of the interventions. Interestingly, articles that predominantly utilized multimedia and a combination of multiple ET ($n = 30$) recorded more positive intervention outcomes. Research studies, such as those conducted by Chowdhury et al. [65] in Bangladesh, which used video-mediated learning to improve farmers' understanding of botanical pesticide usage, and by Bello-Bravo et al. [79], which found an 89% adoption rate when animated agricultural videos was used for the dissemination of postharvest bean storage, clearly demonstrate the effectiveness of multimedia as a reliable tool for promoting the adoption of agricultural technologies. Several studies have examined the effectiveness of mobile apps and smartphones, and four of them reported positive results. One such study was conducted by Dione et al. [76], where the use of interactive voice response (IVR) was found to significantly enhance the knowledge gains of 408 smallholder pig farmers who received biosecurity messages. While the results of the other four were mixed, one study conducted using digital games/simulation also reported a positive outcome which was the study by Dernat et al. [84] where a game-based methodology was found to be very effective in facilitating farmers' collective decision making and continued engagement. Notably, the only article that did not report a positive outcome was a single study that used online/web-based applications. The implications of these findings are that stakeholders in the field of agriculture can collaboratively work together to design a targeted, cost-effective and guaranteed communication channels that could yield greater positive results in the nearest future.

**Table 3. Cross tabulation of educational technology type and effect/impact.**

| Educational Technology | | Effect/Impact of Intervention | | $\chi^2$ |
|---|---|---|---|---|
| | Positive | Non-significant | Mixed | |
| Multimedia | 20 | 1 | 6 | 28.67*** |
| Mixed | 10 | 0 | 5 | |
| Mobile Apps/Smartphones | 4 | 0 | 4 | |
| Digital games/simulation | 1 | 0 | 0 | |
| Online/web-based applications | 0 | 1 | 0 | |

Note *** = p < .001.

In contrast to the analysis on educational technology, the cross-tabulation and Chi-square analysis examining the relationship between the type of agricultural technology provided to farmers and the resulting impact of the intervention did not yield a statistically significant result $\chi^2$ (8, $n$ = 52) = 7.52 ($p$ = .482), as shown in Table 4. Despite the lack of statistical significance, patterns can still be observed between the two variables. Out of the 52 articles, 35 reported a positive outcome, while 15 reported mixed results, regardless of the specific agricultural technology/innovation utilized. These findings suggest that, in the context of agricultural extension, the method of communication or transmission of agricultural information through educational technology may play a more crucial role in determining the overall success of the interventions than the specific agricultural technology employed.

The previous research (44) focused on explaining the process of transferring and adoption of agricultural technology while our study focused on the application/usage of the AT. This study found that simple technology like multimedia served as the most frequently used and video/video-mediated extension served as the most prominent, which is consistent with the previous research [43] stating that technologies that are more complex to comprehend and use have lower rates of adoption. Previous review [44] focused on how one specific type of ET (ICT) affects AT adoption in developing countries while our study investigated diverse kinds of educational technology. Our findings suggested that the use of multimedia as an ET might be due to the characteristics of limited educational level and economic level of farmers in developing countries. It is consistent with previous review [44] indicating that farmers have limited access to resources and infrastructure investments remain low in many developing countries. While these reviews concentrated on measuring the impact of ICT-based agriculture extension programs, our study focused on summarizing the effect/impact of using technologies in agricultural extension with most studies reporting positive outcomes.

**Table 4. Cross tabulation of agricultural technology type and effect/impact of the intervention.**

| Agricultural Technology | | Effect/Impact of Intervention | | $\chi^2$ |
|---|---|---|---|---|
| | Positive | Non-Significant | Mixed | |
| Mixed | 13 | 0 | 8 | 7.52 |
| Pest and disease control | 8 | 0 | 3 | |
| Crop cultivation/harvesting | 6 | 1 | 1 | |
| Product processing | 4 | 0 | 2 | |
| Knowledge/skill/general agricultural education | 4 | 1 | 1 | |

Note: $\chi^2$(8, $n$ = 52) = 7.52, $p$ = .482.

## 6. Conclusion and future directions

In conclusion, this scoping review underscores the critical role of TA in agricultural extension, presenting valuable insights into technology's potential to enhance extension programs and stimulate future research. Maunder's [8] definition of agricultural extension guided this scoping review, emphasizing the characteristics of the service and its potential impact on improving and educating farmers. As explained by Rivera et al. [7], agricultural extension serves as a vital link to increase productivity and efficiency among farmers and researchers, facilitating the sharing of innovations. Technological applications within agricultural extension have the power to transform farming practices [12,13,16].

Through our comprehensive coding, we categorized the TA within agricultural extension into two domains: use of technology/innovation as a factor of production and as an ET. While our study included various agricultural fields, such as agricultural economics, agricultural engineering, animal husbandry, and agronomy, it should be noted that some studies lacked detailed information that could have provided valuable insights into the impact of technology applications on farmers through agricultural extension programs.

Furthermore, this research establishes a foundation for future studies, innovation, and informed practices by identifying areas that warrant further exploration and discovery. The significant increase in research activity in technology applications, particularly after 2016, highlights its growing importance. Advancing the application of technology in agricultural extension contributes to improved agricultural outcomes and sustainable development in farming communities worldwide. Future research on technology applications in agricultural extension should address limitations that may be inherent in the research designs, data collection instruments and the units for the measurement of the intervention outcomes. Future studies should also identify technological effectiveness, delve into mechanisms and contextual factors related to positive outcomes, and aim to support farmers and farm households more effectively.

## Supporting information

**S1 File. Preferred Reporting Items for Systematic Reviews and Meta-Analyses Extension for Scoping Reviews (PRISMA-ScR) checklist.**
(DOCX)

**S2 File. Appendix A- database search strategies.**
(DOCX)

## Author Contributions

**Conceptualization:** Zhihong Xu.

**Data curation:** Zhihong Xu.

**Formal analysis:** Anjorin Ezekiel Adeyemi, Shuai Ma.

**Funding acquisition:** Zhihong Xu.

**Investigation:** Zhihong Xu, Anjorin Ezekiel Adeyemi, Emily Catalan, Shuai Ma, Cristina Guzman.

**Methodology:** Zhihong Xu.

**Project administration:** Zhihong Xu.

**Resources:** Zhihong Xu, Ashlynn Kogut.

**Supervision:** Zhihong Xu.

**Visualization:** Anjorin Ezekiel Adeyemi, Shuai Ma.

**Writing – original draft:** Zhihong Xu, Anjorin Ezekiel Adeyemi, Emily Catalan, Shuai Ma, Cristina Guzman.

**Writing – review & editing:** Zhihong Xu, Anjorin Ezekiel Adeyemi, Ashlynn Kogut.

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
