## [Decision Letter · Decision Letter 0]

14 Aug 2023

PONE-D-23-20077A scoping review on technology applications in agricultural extensionPLOS ONE

Dear Dr. Xu,

Thank you for submitting your manuscript to PLOS ONE. After careful consideration, we feel that it has merit but does not fully meet PLOS ONE’s publication criteria as it currently stands. Therefore, we invite you to submit a revised version of the manuscript that addresses the points raised during the review process.

We look forward to receiving your revised manuscript.

Kind regards,

Mojtaba Kordrostami, Ph.D.

Academic Editor

PLOS ONE

3. Please amend your authorship list in your manuscript file to include authors Zhihong Xu, Anjorin Ezekiel Adeyemi, Emily Catalan, Shuai Ma, Ashlyn Kogut and Cristina Guzman.

4. Please include your tables as part of your main manuscript and remove the individual files. Please note that supplementary tables (should remain/ be uploaded) as separate "supporting information" files.

Additional Editor Comments:

Dear Authors,

I hope this letter finds you well. I would like to extend my gratitude to you for submitting your article to the PLOS ONE journal. Your work has been reviewed by two experts in the field, and I have taken their comments into consideration for the following decision.

Your article presents a compelling exploration of the role of technology in agricultural extension programs. Based on the feedback from the reviewers, the consensus is that your paper is of considerable value to the academic community. The depth of your research, the rigor of your methodology, and the clarity of your writing have been particularly appreciated.

However, both reviewers have pointed out specific areas that could benefit from further clarity, elaboration, or adjustment. These comments are aimed at refining your manuscript to ensure that it provides the most significant value to our readers and the broader academic community.

Reviewer 1:

Reviewer 1 was quite impressed with your manuscript and recommends its acceptance in its current form. They particularly commended the depth of your research, the clarity of writing, and the logical flow of your arguments.

Reviewer 2:

Reviewer 2 provided a detailed breakdown of suggestions and potential areas of improvement. Their feedback spans across several sections of your manuscript, including:

Abstract: Emphasizing the significance of your focus, justifying database choices, providing more context on regional mentions, clarifying the distinction between research methods, and expanding on the impacts and limitations.

Introduction: Incorporating a historical perspective, giving examples of technological integrations, ensuring accurate references, and refining the presentation of objectives.

Literature Review: Streamlining definitions, elaborating on technological impacts, and refining the presentation to avoid redundancy.

Research Questions: Enhancing the specificity of your questions and ensuring that the breadth is maintained throughout the manuscript.

Research Method: Expanding on the search strategy, clarifying methodological choices, incorporating a PRISMA flow diagram, and reflecting on challenges faced during research.

Results and Discussion: Providing insights on publication platforms, discussing regional research discrepancies, interpreting statistical results, and drawing comparisons with other literature.

In light of the above, I am returning your manuscript with a decision of "Revise". I believe that by addressing the reviewers' comments, your manuscript can be further enhanced, making it an even more valuable contribution to our journal and the field at large.

Please ensure that you address each point raised by the reviewers. Upon resubmission, kindly include a detailed point-by-point response indicating how you have addressed the reviewers' comments or provide a rationale if certain suggestions were not incorporated.

We appreciate the time and effort you have put into your research and manuscript. I hope you find the reviewers' feedback constructive. I am looking forward to receiving your revised manuscript.

Warm regards,

Mojtaba Kordrostami

Editor

PLOS ONE Journal

Reviewers' comments:

Reviewer's Responses to Questions

**Comments to the Author**

1. Is the manuscript technically sound, and do the data support the conclusions?

Reviewer #1: Yes

Reviewer #2: Yes

2. Has the statistical analysis been performed appropriately and rigorously? 

Reviewer #1: Yes

Reviewer #2: No

3. Have the authors made all data underlying the findings in their manuscript fully available?

Reviewer #1: Yes

Reviewer #2: Yes

4. Is the manuscript presented in an intelligible fashion and written in standard English?

Reviewer #1: Yes

Reviewer #2: No

5. Review Comments to the Author

Reviewer #1: This is a very interesting article. The author has delved deep into the subject matter, presenting well-researched insights and thoughtful arguments. The clarity of writing and logical flow make it an engaging read. The article effectively captures the reader's attention from the beginning till the end. The supporting evidence and references add credibility to the claims made. Overall, it's a valuable contribution to the field. I highly recommend accepting it in the current form. Great work!

Reviewer #2: Dear Editor

Please find my comment below:

Abstract

General Comments

The abstract presented offers an in-depth scoping review of technology's role in agricultural extension programs. The approach is comprehensive, and the narrative effectively captures the integration of both agricultural and educational technologies. The use of multiple databases for sourcing articles provides a robust foundation for the findings. The structured presentation of findings is also commendable.

However, certain aspects of the abstract would benefit from added clarity or additional information. Specific details and clearer articulation in some areas would enhance the reader's understanding and make the abstract even more impactful.

Specific Comments

1. The abstract briefly touches upon the lack of previous reviews on the impact of technology in agricultural extension. It would be beneficial if the authors briefly indicate why this particular focus is of significance.

2. Justifying the choice of the five databases, or mentioning if these are the most prominent databases in this field, would enhance the credibility of the study.

3. The mention of India and Africa requires more context. It would be useful to know if this observation indicates a trend or if it merely represents the scope of available literature.

4. The distinction between the quantitative research method being the most employed and the mixed methods being the most used data collection approach might be confusing. It would be helpful if the authors could provide a brief explanation or example of this distinction.

5. While the most widely used educational technology is mentioned, the abstract could benefit from highlighting a few of the most common agricultural technologies that appeared in the reviewed studies.

6. The statement that the impacts were "mostly mixed" requires further specificity. Providing a brief example or elaborating on what areas showed positive or negative impacts would be beneficial.

7. It's commendable to acknowledge potential limitations. A brief mention of one or two key limitations would be insightful.

8. The abstract concludes with an emphasis on gaps in the literature. Mentioning one or two primary gaps or areas for future research would provide readers with a clear takeaway.

Introduction

General Comments

The introduction offers a clear context and rationale for the importance of integrating technology into agricultural extension programs. The progression from the significance of technology in enhancing extension programs to the purpose of the scoping review is logical. The emphasis on the potential benefits for policymakers, researchers, and practitioners provides a broad perspective on the review's relevance.

However, some areas could benefit from further elaboration, and the structure might be enhanced to offer a more concise and direct presentation of the main points.

Specific Comments

1. The introduction starts strongly by emphasizing the importance of agricultural extension programs. However, it could benefit from a brief mention of the historical or traditional methods of agricultural extension for context.

2. While the importance of technology in agricultural extension is emphasized, it would be beneficial to provide examples or categories of such technologies. This would offer readers a clearer picture of what technological integrations are being discussed.

3. The references (1) and (2) are placeholders. In the final manuscript, it would be crucial to ensure that these references are accurately representing the claims made.

4. The statement about shedding light on the "current state of research" and mapping the field is clear. However, distinguishing between the broader goals of the review and the specific objectives could provide more clarity.

5. The mention of policymakers, researchers, and practitioners is appropriate. Still, it might be enhanced by briefly discussing the specific challenges or questions each of these groups faces that the review can address.

6. The final part of the introduction discusses the research's aims to lay a foundation for future studies. While this is a strong ending, it might be enhanced by presenting a more concise summary of the intended contributions and outcomes of the review.

Literature Review

General Comment

The literature review offers a comprehensive overview of the integration of technology in agricultural extension programs. The authors have meticulously categorized the research into the historical perspectives of agricultural extension, the role of technology in agriculture, and the intersection of both. The reference to prior studies and the identification of gaps in existing literature lend robustness to the review.

However, some areas could benefit from further clarity, and the structure might be enhanced to offer a more concise presentation of the main points.

Specific Comments

1. Agricultural Extension Definitions: The various definitions of agricultural extension provided are comprehensive. However, the transition to the definition that the review aligns with could be smoother. Perhaps a brief rationale for choosing Maunder’s definition would be beneficial.

2. Technological Integration: The distinction between agricultural technology as a component of production and as an educational tool is clear. Yet, more explicit connections between the tools and their practical impacts would enhance understanding. For instance, how do drones or IoT directly influence agricultural extension?

3. Previous Studies and Research Gap: While the section thoroughly identifies gaps in existing research, it could benefit from a more streamlined presentation. The repeated mention of "technology application in agricultural extension" and the emphasis on the review's unique approach can be condensed to avoid redundancy.

4. Citation and Referencing: The placeholders for references are well-placed, providing a strong foundation for the claims made. In the final manuscript, ensuring that these references are comprehensive and up-to-date will be critical.

5. Relevance of Previous Studies: The review does well to distinguish itself from the works of Altalb et al. and Aker. However, a brief mention of why these studies are particularly relevant or how they shaped the current review's approach might provide more context.

6. The literature review concludes with a forward-looking statement about enhancing productivity and bridging divides. This is effective but could be enhanced with a brief mention of the expected outcomes or implications of the scoping review.

Research Questions

General Comments

The research questions section offers a structured breakdown of the areas that the scoping review aims to address. The categorization into substantive features, methodological features, and characteristics of technology application provides a clear roadmap of the study's approach. The questions themselves are well-formulated and adequately detailed, promising a comprehensive exploration of the topic.

However, certain areas could benefit from further specificity or clarity to ensure that the subsequent sections of the manuscript align seamlessly with these guiding questions.

Specific Comments

1. While the query about publication information is clear, it might be helpful to specify what particular publication information is of interest (e.g., publisher, year, journal name). Additionally, the inclusion of "agricultural field" is relevant, but the term might benefit from elaboration or examples for clarity.

2. The question is comprehensive in covering research methods, data collection approaches, and sample size. However, it might be enhanced by adding inquiries about potential research biases, limitations, or challenges identified in the included studies.

3. The distinction between educational technology and agricultural technology is clear and aligns with the literature review. Yet, the question might benefit from an exploration of the integration or interaction of these technologies. For instance, how does the use of educational technology influence the adoption or effectiveness of agricultural technology?

4. The query about the "overall effect of technology on agricultural extension" is broad. It would be beneficial to specify if this effect is being measured in terms of productivity, knowledge transfer, farmer satisfaction, or any other specific metrics.

5. The research questions set a broad scope for the review. Ensuring that this breadth is maintained throughout the manuscript will be crucial, especially in the results and discussion sections.

Research Method

General Comments

The research method section is comprehensive, providing detailed insights into the procedures followed in the scoping review. The use of multiple databases, clearly defined inclusion and exclusion criteria, and a structured coding scheme reflects the systematic approach the authors have taken. The use of PRISMA flow and the description of inter-rater reliability further emphasize the rigor with which the study has been conducted.

While the overall methodology appears robust, certain areas could benefit from further clarity or elaboration to ensure the methodological choices are entirely transparent and replicable.

Specific Comments

o It is commendable that the authors have provided the date of the search to ensure the recency of the data.

o While the Appendix A contains the full search strategy for CAB Abstracts, the modifications made to fit other databases would be useful for replication. It would be beneficial to briefly describe or provide these modified strategies in an additional appendix.

o The criteria are well-defined and comprehensive. However, the delineation between what qualifies as an educational technology versus agricultural technology might benefit from additional examples or elaboration.

o It would be helpful to know why the authors chose the specific date range of January 1, 2000, to November 1, 2022. While technological advancements since 2000 are mentioned, a brief rationale for this specific range could enhance clarity.

o The coding scheme is extensive and well-structured. However, the categorization of agricultural field/enterprise could benefit from a more exhaustive list or examples, given that only a few are mentioned.

o Under the section on methodological features, while the grouping of research methods is clear, a brief rationale for these groupings (especially what constitutes mixed methods) would be useful.

o The distinction between educational technology and agricultural technology/innovation is clear. However, the list of technologies and their sub-categories might benefit from further examples or references to ensure clarity.

o The PRISMA flow diagram, while mentioned, is not provided within the section. If feasible, it would be helpful to include this diagram directly within the manuscript or provide a clearer direction to its location.

o The inter-rater reliability is commendably high, but a brief discussion on how discrepancies were resolved (other than the first author acting as arbiter) would provide additional transparency.

o It might be helpful to provide a brief overview of the descriptive statistical analyses planned or executed to address the research questions.

o The section could benefit from a brief discussion or reflection on any anticipated or encountered challenges during the research method, especially during data collection or coding.

Results and Discussion

General Comments

The Results and Discussion section is extensively detailed, covering a wide range of aspects concerning the substantive features, methodological features, and characteristics of educational technology in agricultural extension. The use of figures, tables, and statistical analyses enhances the rigor and depth of the presented findings. The section is well-organized, with clear sub-sections that aid in understanding the progression of results.

However, there are areas that could benefit from further elaboration or explanation to ensure clarity and completeness.

Specific Comments

o The graphical representation of publication distribution (Fig 2) and the breakdown of journals, conferences, and policy papers (Table 2) provide a clear overview of the landscape of research in the field. It would be beneficial to include comments or insights on the top journals or platforms publishing in this area.

o The distribution by country and region paints a clear picture of where the research is focused. Some insight into the potential reasons behind the lack of research from certain regions, like Europe or Australia/Oceania, beyond what is provided, might enhance the discussion.

o The breakdown of research methods, data collection approaches, inferential statistics, and sample size units are comprehensive. It would be interesting to see a further discussion on the implications or reasons behind the prevalent use of quantitative methods over qualitative ones.

o The discussion on sample size units and the categorization based on the number of participants adds depth to the results. However, a brief discussion on the implications of these findings for future research would enhance this section.

o The breakdown of different types of educational technologies and agricultural technologies is detailed and clear. It would be helpful to delve deeper into the reasons behind the prevalent use of certain technologies over others.

o The findings on the intervention characteristics of technology are insightful. The relationship between the duration, intensity, and outcomes of interventions could benefit from further exploration.

o The cross-tabulation analyses and the Chi-square tests add depth to the results, providing a clear understanding of the relationships between variables. However, some additional interpretation of these results in the context of the broader research landscape would be useful.

o The section might benefit from a summarization of the main findings and their implications for both researchers and practitioners in the field.

o While the section is quite detailed, it would be beneficial to see more connections or comparisons with other studies or literature in the field, providing a broader context for the presented findings.

6. PLOS authors have the option to publish the peer review history of their article (what does this mean?). If published, this will include your full peer review and any attached files.

Reviewer #1: **Yes: **Cristiano Matos

Reviewer #2: No

---

## [Author Response · Author response to Decision Letter 0]

28 Sep 2023

Dear reviewers,

We are delighted to revise our manuscript based on the excellent feedback from the reviewers and you. We believe the suggestions and corresponding revisions have significantly improved our research, findings, and manuscript. We responded to the comments one by one and highlighted what we revised in the manuscript. We also created a comments and response table to explain how we addressed all of the comments. We will be happy to receive any additional comments and make revisions if necessary. Thanks again for the opportunity to revise and resubmit.

Detailed information can be found in our Comments and Response table and the revised manuscript. 

Sincerely,

Zhihong Xu

---

## [Decision Letter · Decision Letter 1]

2 Oct 2023

A scoping review on technology applications in agricultural extension

PONE-D-23-20077R1

Dear Dr. Xu,

We’re pleased to inform you that your manuscript has been judged scientifically suitable for publication and will be formally accepted for publication once it meets all outstanding technical requirements.

Kind regards,

Mojtaba Kordrostami, Ph.D.

Academic Editor

PLOS ONE

Additional Editor Comments (optional):

The manuscript can be accepted now.

Reviewers' comments:

Reviewer's Responses to Questions

**Comments to the Author**

1. If the authors have adequately addressed your comments raised in a previous round of review and you feel that this manuscript is now acceptable for publication, you may indicate that here to bypass the “Comments to the Author” section, enter your conflict of interest statement in the “Confidential to Editor” section, and submit your "Accept" recommendation.

Reviewer #2: All comments have been addressed

2. Is the manuscript technically sound, and do the data support the conclusions?

Reviewer #2: Yes

3. Has the statistical analysis been performed appropriately and rigorously? 

Reviewer #2: Yes

4. Have the authors made all data underlying the findings in their manuscript fully available?

Reviewer #2: Yes

5. Is the manuscript presented in an intelligible fashion and written in standard English?

Reviewer #2: Yes

6. Review Comments to the Author

Reviewer #2: DEAR EDITOR

THE MANUSCRIPT IS IMPROVED SIGNIFICANTLY AND CAN BE ACCEPTED NOW.

It can be accepted now.

Regards

7. PLOS authors have the option to publish the peer review history of their article (what does this mean?). If published, this will include your full peer review and any attached files.

Reviewer #2: No

---

## [Editor Report · Acceptance letter]

27 Oct 2023

PONE-D-23-20077R1 

A scoping review on technology applications in agricultural extension 

Dear Dr. Xu:

I'm pleased to inform you that your manuscript has been deemed suitable for publication in PLOS ONE. Congratulations! Your manuscript is now with our production department. 

Kind regards, 

on behalf of

Dr. Mojtaba Kordrostami 

Academic Editor

PLOS ONE